**Subject Category:**
Biology (whole organism)

evolution/taxonomy and systematics/ palaeontology

phylogenetics, palaeontology, Mammalia, Geomyoidea, Sciuromorpha

**Author for correspondence:**
Robert J. Asher
e-mail: r.asher@zoo.cam.ac.uk

# Congruence, fossils and the evolutionary tree of rodents and lagomorphs

Robert J. Asher[1], Martin R. Smith[2], Aime Rankin[1] and Robert J. Emry[3]

[1]Department of Zoology, University of Cambridge, Cambridge, UK
[2]Department of Earth Sciences, University of Durham, Durham, UK
[3]Department of Paleobiology, Smithsonian Institution, Washington, DC, USA

RJA, 0000-0002-4434-9074; MRS, 0000-0001-5660-1727

Given an evolutionary process, we expect distinct categories of heritable data, sampled in ever larger amounts, to converge on a single tree of historical relationships. We tested this assertion by undertaking phylogenetic analyses of a new morphology-DNA dataset for mammals, focusing on Glires and including the oldest known skeletons of geomyoid and *Ischyromys* rodents. Our results support geomyoids in the mouse-related clade (Myomorpha) and a ricochetal locomotor pattern for the common ancestor of geomyoid rodents. They also support *Ischyromys* in the squirrel-related clade (Sciuromorpha) and the evolution of sciurids and *Aplodontia* from extinct, 'protrogomorph'-grade rodents. Moreover, ever larger samples of characters from our dataset increased congruence with an independent, well-corroborated tree. Addition of morphology from fossils increased congruence to a greater extent than addition of morphology from extant taxa, consistent with fossils' temporal proximity to the common ancestors of living species, reflecting the historical, phylogenetic signal present in our data, particularly in morphological characters from fossils. Our results support the widely held but poorly tested intuition that fossils resemble the common ancestors shared by living species, and that fossilizable hard tissues (i.e. bones and teeth) help to reconstruct the evolutionary tree of life.

## 1. Introduction

The theory of biological evolution leads to many hypotheses about the history of life. Given an evolutionary mechanism in which heritable variation yields biodiversity over geological time, we expect patterns of relationship derived from independent sources of data to exhibit similarity, and for that similarity to increase as more data are analysed with methods capable of measuring historical signal [1,2]. Data typically used

for phylogenetic purposes include genotypes and phenotypes, and many subcategories within each, including (but not limited to) mitochondrial and nuclear DNA exons, introns, retrotransposons and short interspersed nuclear elements (SINEs), rare genomic changes (RGCs), ultraconserved elements (UCEs), patterns of insertions and deletions (indels), as well as anatomical patterns of development, soft tissues, bones and teeth. Fossilization almost always reduces data to hard tissues only, but we still expect that overall, increasing amounts of such phenotypic data should yield patterns of relationship that converge on those based on soft tissues and genotype. Tests of evolution as the primary mechanism behind biodiversity are numerous and compelling [3]. However, the extent to which anatomical data in general, and fossilizable hard tissues in particular, help or hinder phylogenetic reconstruction remains contentious. Estimates of morphology's phylogenetic information content range from limited [4] to substantial [5], and that of morphological hard tissues from biased [6] to statistically consistent [7], at least for some groups.

Here, we use biologically and palaeontologically well-documented rodents and lagomorphs (collectively known as Glires) to test predictions derived from evolutionary theory. We assembled a morphological matrix sampling extant and fossil mammals, including the oldest and most complete skeletons yet known of an Eocene geomyoid (figure 1, *Heliscomys* [8]) and 'ischyromyid' (figure 2, *Ischyromys* [9]), as well as previously undescribed cranial fossils of *Douglassciurus* [10]. We combined hard tissue data from 42 fossils with morphological and molecular characters from 60 extant genera in order to measure, firstly, congruence of topologies derived from increasingly large samples from our dataset to a well-corroborated tree (figure 3), derived from SINEs [11], microRNAs ([12]: their fig. S5), introns [13], and rare genomic changes [14], data which played no role in our own original dataset. This tree is furthermore consistent with an analysis of ultraconserved elements from over 3700 nuclear loci [15], which shows less than 0.002% (8/3700) overlap with the eight nuclear genes in our original alignment. By 'well-corroborated', we mean that one or an extremely small number of mutually consistent topologies, out of an astronomically huge number of possibilities, result from phylogenetic optimality criteria applied to these datasets. Secondly, we measure the phylogenetic impact of fossils, extant taxa and hypothetical ancestors [16] on congruence with this well-corroborated topology of extant taxa. We hypothesize that if evolution left a historical signal in bones and teeth, then ever-increasing samples from our dataset should converge on the well-corroborated species tree of mammals in general and Glires in particular.

## 2. Material and methods

We combined 14 genes (eight nuclear, six mitochondrial) with their patterns of insertions and deletions (indels) in a matrix for 60 living genera, yielding an alignment of 15 407 nucleotides and 188 binary indels. Our morphology partition consists of 219 hard tissue characters coded for those 60 taxa plus another 42 fossils. We coded all taxa for morphological characters [17,18] and incorporated further improvements based on Wible [19], von Koenigswald [20,21] and digital reconstructions from new specimens based on CT scans taken primarily at the Cambridge Biotomography Centre (UK) and rendered in 3D with Drishti 2.6.4 [22]. Graphic documentation of all character states are available in project 2769 on morphobank.org [23]. DNA+indel characters were unavailable for fossils. Species were chosen to maximize overlap of morphological data with available DNA sequences. Our sample of 60 extant mammals included a core of 41 genera with known phylogenetic affinities based on SINEs [11], microRNAs [12], introns [13] and RGCs [14], all of which are independent of the novel morphological and DNA data in our dataset and are furthermore consistent with analysis of UCEs noted above [15]. The consensus of these studies defines a well-corroborated tree of Mammalia, focusing on rodents and lagomorphs (figure 3). Electronic supplementary material provides further details on taxon and character sampling, DNA alignment, modelling and character partitioning, phylogenetic search strategies, electronic supplementary material, figures S1–S7 and tables S1–S5. Our morphological matrix and DNA+indel alignments are available in electronic supplementary material, appendix S1; electronic supplementary material, appendix S2 provides a summary of character coding edits relative to past phylogenetic analyses of fossils and morphology (e.g. [17,18]); electronic supplementary material, appendix S3 provides species-level taxonomy, museum accession numbers for specimens used to code morphological data and GenBank accession numbers; electronic supplementary material, appendix S4 provides our optimal topologies in nexus format. These electronic supplementary material appendices are available at datadrad.org: https://doi.org/10.5061/dryad.3840vd7 [24]. Electronic supplementary material, tables S1 and S4 summarize our DNA partitioning scheme, model choice and indices of convergence for our Bayesian analyses.

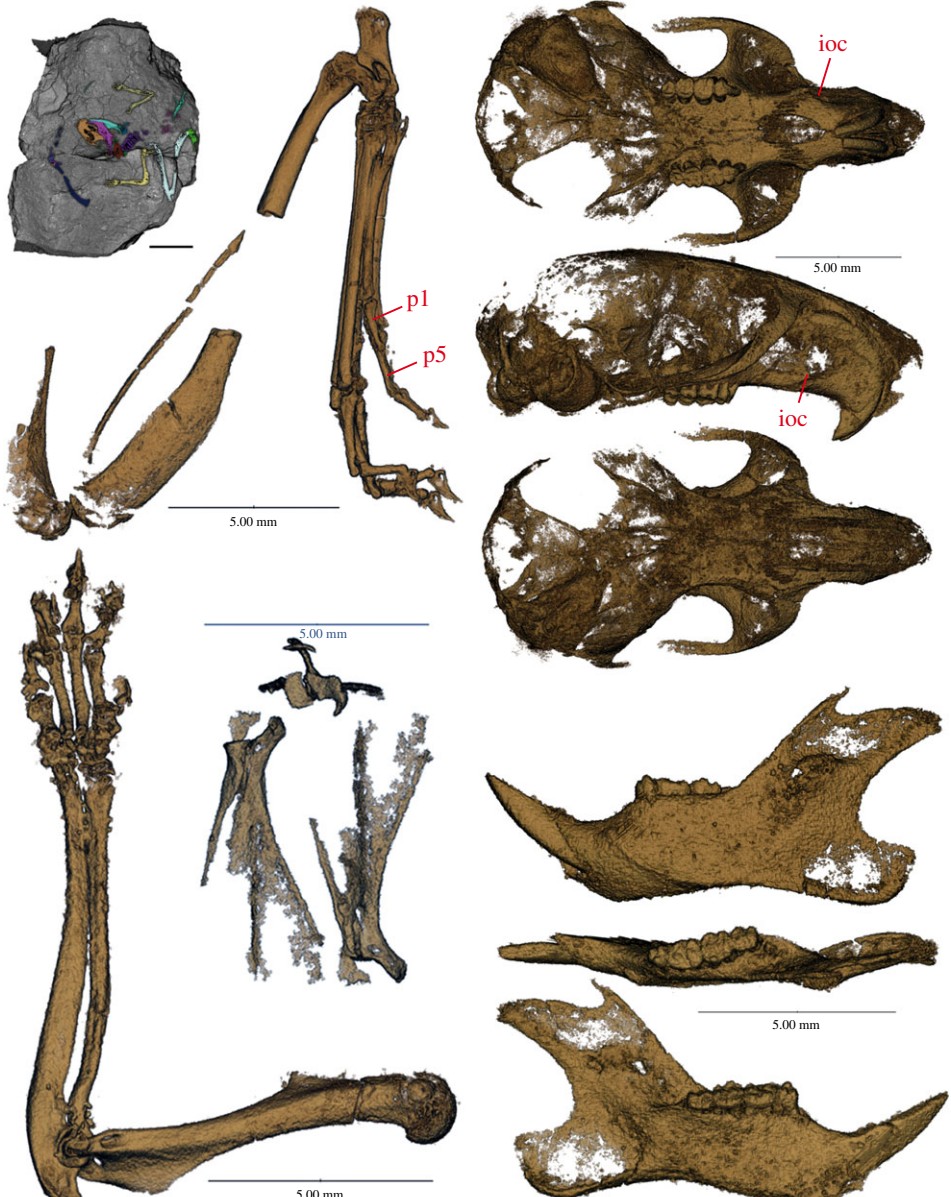

**Figure 1.** *Heliscomys ostranderi* (USNM PAL 720183) from the Chadronian (latest Eocene) of Jenny's Pocket, Flagstaff Rim, Wyoming. Virtually dissected, CT-scanned images are (clockwise from upper left) block in which specimen was found with elements coloured therein; left hindlimb in medial view; skull in ventral, lateral and dorsal views; left jaw in lateral, occlusal and medial views; right forelimb in medial view; partial scapula in dorsal, ventral and lateral views. ioc; anterior entrance of infraorbital canal; p1, proximal phalanx of pedal digit I; p5, proximal phalanx of pedal digit V. Scale bars, 10 mm at top left, 5 mm elsewhere.

Beck & Baillie [16] used a novel method to incorporate data from living taxa in studies of fossils where genomic data are generally unavailable. Using the topology from Meredith *et al.* [25], they optimized morphological characters at internal nodes and re-included these nodes as taxa for additional phylogenetic analysis. Inclusion of such hypothetical ancestors resulted in a substantial increase in congruence of topologies generated by morphology alone with that generated by the 26 nuclear gene fragments from Meredith *et al.* [25]. Ancestors inferred using maximum parsimony (MP) were 'somewhat more successful in recovering' [16] well-corroborated clades compared to likelihood-inferred ancestors. They assumed the phylogeny of Meredith *et al.* [25] to be accurate and regarded hypothetical common ancestors inferred from that phylogeny to represent a 'best case scenario' in palaeontology; i.e. such ancestors represent hypotheses of the morphology of real common ancestors from which modern clades have evolved. Their study was not a test of evolution *per se*, or a test of the topology proposed by Meredith *et al.* [25], but it did provide a means by which to increase congruence of topologies derived from fossils with topologies based on living taxa sampled for DNA. The method of Beck &

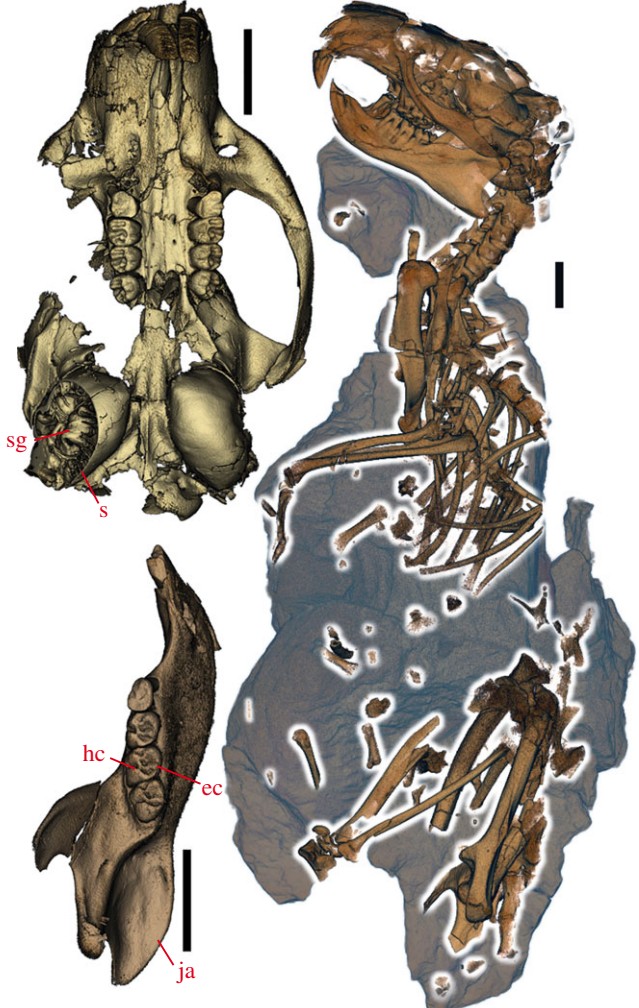

**Figure 2.** *Ischyromys* sp. USNM PAL 617532 from the Duchesnean (late Eocene) of West Canyon Creek, Wyoming. Virtually dissected skeleton in original matrix (right), skull in ventral view with R & L dP4-M3 and cutaway into right auditory bulla (top left), left jaw in occlusal view with dp4-m3 (bottom left). ec, entoconid; hc, hypoconid; ja, angle of the jaw; s, intrabullar septae; sg, stapedial groove. Scale bars, 10 mm.

Baillie[16] thus provides a means to include the phylogenetic information from modern taxa in an analysis of fossils, without directly having to sample non-fossilizable data, such as DNA.

Here, we used the 'describetrees /xout=internal' command in PAUP 4.0a [26] to estimate states for our 219 morphological characters for each of the 29 internal nodes in the well-corroborated topology (figure 3). This tree was based on data noted above [11–14], independent of the DNA and morphological data used in our own phylogenetic analyses. We generated character states assuming both accelerated ('acctran') and delayed ('deltran') transformations and added these hypothetical terminals to our morphology matrix, and manually ensured that inapplicable character optimizations were coded as such in our hypothetical ancestors. Acctran and deltran generated similar results (electronic supplementary material, figure S1 and appendix S1). We acknowledge that optimizing characters on a topology with polytomies (as evident in our well-corroborated tree, figure 3) departs from the uncertainty generally associated with non-bifurcating nodes [27].

We relied primarily on two quartet metrics to quantify the similarity (amount of information in common) between the well-corroborated tree (figure 3) and each test tree: (i) a symmetric difference measure [28] normalized against the maximum information attainable (function QuartetDivergence in the Quartet R package [29]) and (ii) a shared information measure, which calculated the number of shared quartets normalized against the number of quartets resolved in the independent tree, using the function SharedQuartetStatus in the Quartet package. Where taxon samples differed, taxa not held in common were pruned using drop.tip (in ape [30]) before calculating consensus trees (R script available in electronic supplementary material, table S5). We also made comparisons using two further metrics: (i) Robinson-Foulds

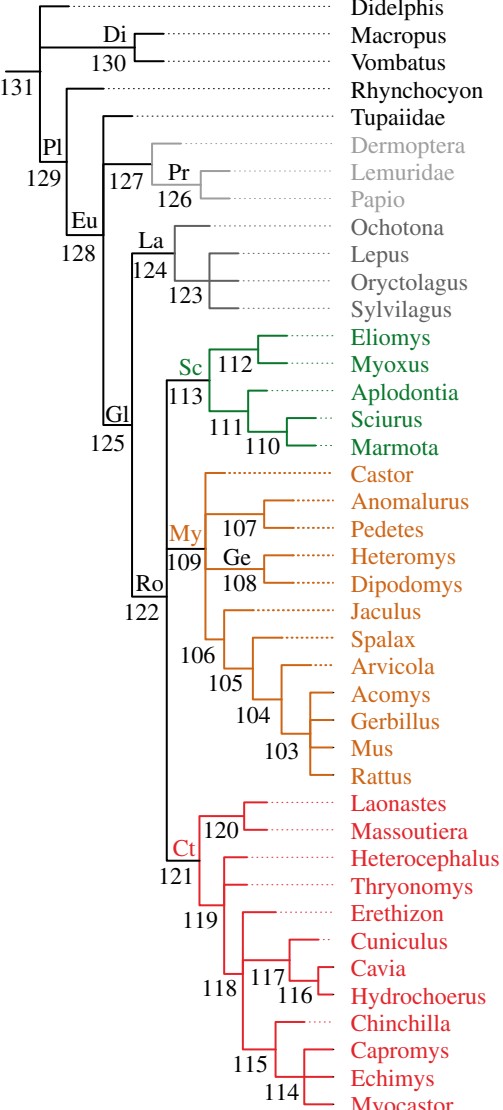

**Figure 3.** Well-corroborated mammalian phylogenetic tree, focusing on Glires, based on consensus from SINEs [11], microRNAs [12], introns [13] and RGCs [14]. Ct, Ctenohystrica; Di, Diprotodontia; Eu, Euarchontoglires; Ge, Geomyoidea; Gl, Glires; La, Lagomorpha; My, Myomorpha; Pl, Placentalia; Pr, Primates; Ro, Rodentia; Sc, Sciuromorpha. Primates-Dermoptera are shown in light grey, Lagomorpha in dark grey, Sciuromorpha in green, Myomorpha in orange and Ctenohystrica in red. Numbers represent hypothetical ancestors onto which morphological characters were optimized (see electronic supplementary material, appendix S1). Branch lengths are arbitrary.

distance (RF.dist in the Phangorn package [31]) scaled by proportion resolved, RF.dist (testtree,reftree)/ (Nnode(testtree)/length(testtree$tip.label)-1), and (ii) counts of shared partitions using either Mesquite [32] or with prop.clades in the ape package, sum(prop.clades(testtree,reftree)). All calculations of congruence are relative to the well-corroborated tree (figure 3) and derive from Newick-formatted, strict consensus topologies rooted on *Didelphis*.

We evaluated our hypotheses on the phylogenetic information content of hard-tissue data and fossils through a series of resampling experiments. These iteratively increased the number of sampled DNA characters with and without morphology, and also iteratively increased the number of sampled fossils, hypothetical ancestors and sub-sampled living taxa, as detailed below.

# 3. Results

Our analysis supports Glires and three major clades within crown Rodentia: Sciuromorpha, Myomorpha, and Ctenohystrica; this nomenclature is justified in electronic supplementary material. Topologies

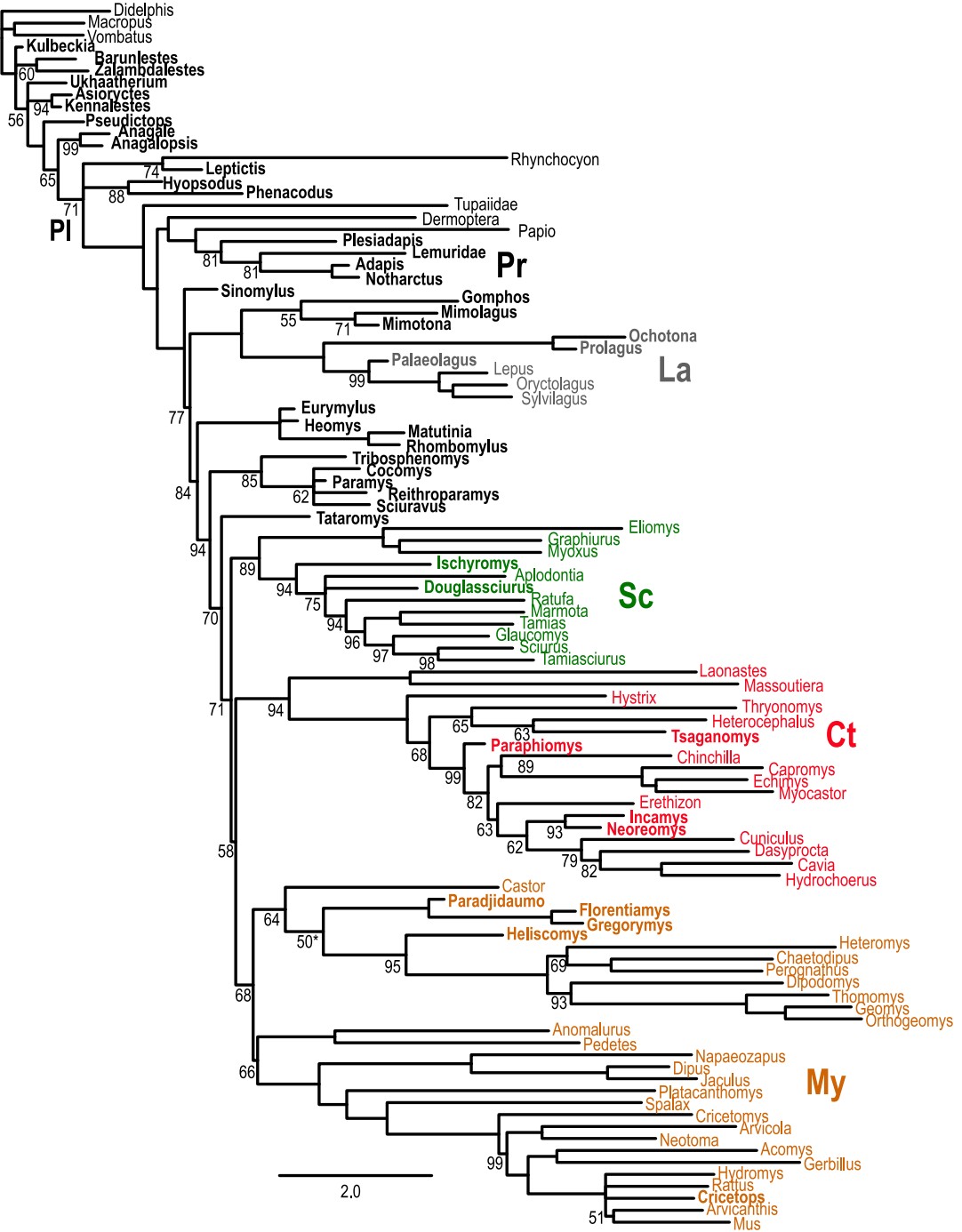

**Figure 4.** Majority rule consensus of 17 500 post-burnin (50% of 35G generations sampling every 1000) Bayesian topologies based on DNA+indel+morphology dataset with posterior probabilities shown as percentages adjacent to each node. Resolved, internal nodes without a number have a posterior probability of 1.0 (=100%). Bold indicates fossils; branch lengths within Placentalia (demarcated with 'PL') correspond to scale at bottom. Colours and high-level taxon abbreviations are as in figure 3. Burn-in values defined in electronic supplementary material, table S4 yield the same topology and similar support values, except for collapsing the '50' node within the mouse-related clade identified with the asterisk.

derived from Bayesian (figure 4) and parsimony (MP; electronic supplementary material, figure S2) searches applied to our combined dataset, with or without fossil taxa, are highly congruent with the independent, well-corroborated tree (figure 3), and to only a slightly lesser extent to previous palaeontological estimates based on dental data [33]. Bayesian analysis (figure 4) shows higher congruence by reconstructing primates together to the exclusion of Dermoptera, whereas most MP analyses with DNA+indels favour Dermoptera-*Papio* to the exclusion of strepsirhines.

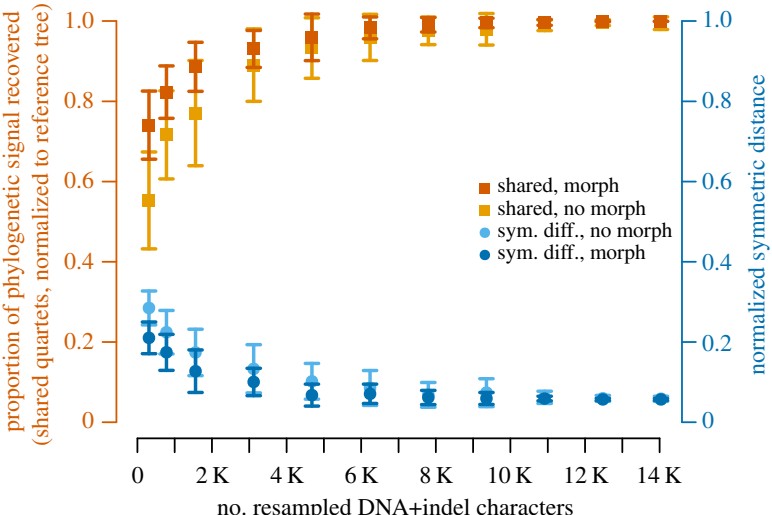

**Figure 5.** Proportion of shared quartets (orange squares), and normalized symmetric difference (blue circles), between strict-consensus MP topologies of extant taxa relative to the well-corroborated tree (figure 3). X-axis shows 2%, 5%, 10%, 20%, . . . 90% resamples of 15 596 DNA+indel characters (each consisting of 25 sets of randomly chosen sites) with (dark shades) and without (light shades) 219 morphological characters. Y-axis error bars show one standard deviation on either side of the mean (polygons).

In order to explore the extent to which increasingly large subsamples of our dataset converged on the well-corroborated tree of Mammalia, we undertook a series of resampling experiments that gradually increased the number of sites from our alignment used in phylogenetic analyses. We ran 50 MP analyses each for 11 different levels of sampling, from 2% to 90% coverage of our alignment. For each level we ran 25 MP analyses with and 25 without morphology and calculated the mean and standard deviation of similarity metrics relative to the well-corroborated tree (figure 3). Both metrics exhibit a clear asymptote towards congruence (figure 5). As more DNA+indel sites are added, the amount of conflict falls and information in common increases. This is limited by the inability of MP analyses without morphology to recover monophyly of primates relative to Dermoptera, a limitation which is overcome in some instances with the addition of morphology and in other studies which have a better sample of primates [7].

Our morphological dataset consists of 219 characters and cannot be subsampled as extensively as our DNA+indel alignment. However, it can be subsampled by taxon (below), and we can also compare how our subsampled DNA+indel analyses perform with and without morphological data. For each of our 11 categories of resampled data, addition of morphology decreases conflict and increases information in common between a given test tree and the well-corroborated tree. This improvement can be substantial; randomly subsampled DNA+indel analyses recovered significantly fewer shared quartets with the well-corroborated tree (figure 3) than those same analyses combined with the 219 morphological characters, although significance varies depending on the test used (electronic supplementary material, table S2). Mean congruence values improved with the addition of morphological data for all of our subsampled analyses (figure 5), and it is only with the addition of morphology that some MP analyses recovered (for example) a monophyletic primate to the exclusion of Dermoptera. This well-corroborated signal appears in our optimal Bayesian trees (figure 4; electronic supplementary material, figure S3A) but not in the largest MP analyses (electronic supplementary material, figures S2 and S3B).

## 3.1. Fossils and hypothetical ancestors improve congruence

Gradually increasing the number of sampled fossils and hypothetical ancestors to our dataset of 60 living taxa, using morphological characters alone, increased similarity and reduced conflict (figure 6) relative to the well-corroborated tree. The correlation between increased congruence and increased sampling of taxa known for morphology was highly significant both for fossils and hypothetical ancestors. In agreement with Beck & Baillie [16], addition of hypothetical ancestors was particularly effective and exhibited increased congruence with a higher slope (i.e. more congruence per added taxon) compared to fossils (figure 6). Although these morphology-only analyses exhibited increased congruence with added fossils and hypothetical ancestors, they still show more conflict than topologies derived from the

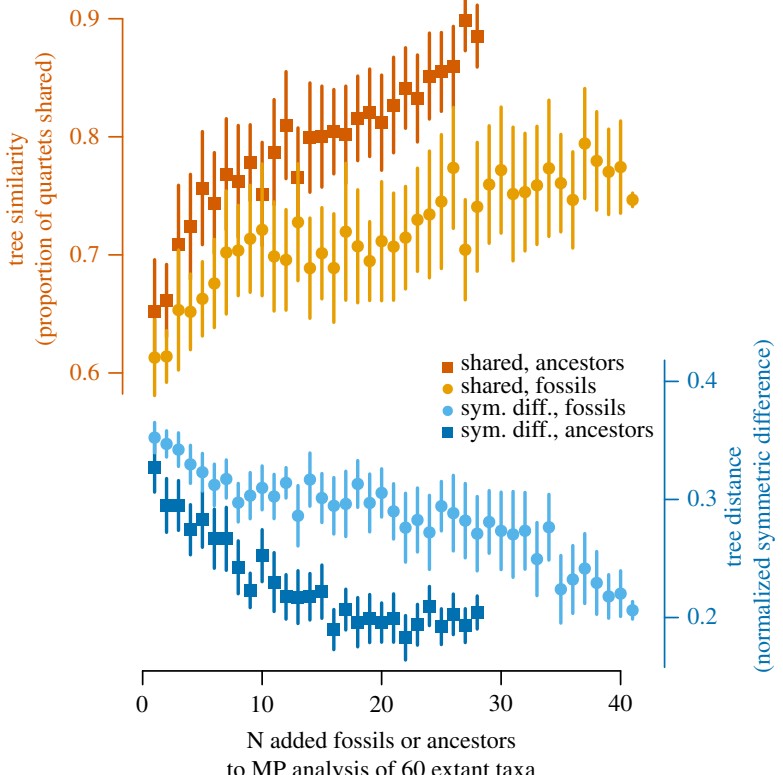

**Figure 6.** Shared quartets (top) and normalized symmetric difference (bottom) between the well-corroborated tree and strict consensuses generated by equal weights MP analysis of morphological data from 60 extant taxa, with 25 analyses per N added fossils (circles, pale) or hypothetical ancestors (squares, dark) on X-axis. Vertical error bars represent one standard deviation on either side of the mean (polygons).

combined or DNA+indel datasets. However, unlike the MP subsampled analyses of our DNA+indel dataset (figure 5), the slope of increased similarity and reduced conflict with the addition of fossils does not appear to asymptotically approach a limit (figure 6), implying that, at least for our dataset, congruence would continue to increase with further addition of fossils.

In order to compare the addition of fossils and hypothetical ancestors to addition of living taxa, we subsampled from our set of 60 extant taxa. The core of 41 taxa in our sample with well corroborated affinities (figure 3) allowed us to divide our dataset into those 41 taxa plus 18 unique sets of one to 18 additional extant taxa (given that there is only one set of all 60 extant taxa, 41 + 19, in our dataset). We therefore subsampled random combinations (18 iterations each) of up to 18 other extant taxa, hypothetical ancestors and fossils in order to further explore congruence with the well-corroborated tree. Again, this smaller series of added hypothetical ancestors significantly decreased conflict in terms of normalized symmetric differences (figure 7); addition of fossils did not monotonically do so, but addition of 18 distinct sets of 18 fossils each did result in a higher overall mean number of shared partitions than 18 distinct sets of 1–17 added fossils (electronic supplementary material, figure S4). In contrast, successive addition of randomized sets of up to 18 other extant taxa to the 41-taxon set from our well-corroborated tree significantly increased conflict as measured by either normalized symmetric difference (figure 7) or raw number of shared partitions (electronic supplementary material, figure S4) with the well-corroborated tree. Randomly added living taxa, not in the 41 taxa in the well-corroborated tree (figure 3), include both relatively long- (e.g. *Cricetomys, Hystrix, Napaeozapus, Platacanthomys* and *Ratufa*) and short- (e.g. *Arvicanthis, Dipus, Geomys* and *Tamiasciurus*) branched taxa, according to our optimal Bayesian tree (figure 4).

Although morphological characters significantly improved congruence when combined with our DNA+indel dataset, the extent to which they do so in isolation depends greatly on taxon sample (electronic supplementary material, figures S5 and S7), character weighting (electronic supplementary material, figures S6 and S7) and optimality criterion (electronic supplementary material, figures S7 and table S3). Equally weighted MP applied to extant taxa recovers only six out of 28 possible well-corroborated groups (calculated with 'shared partitions' in Mesquite [32]). This increases to 12 with a

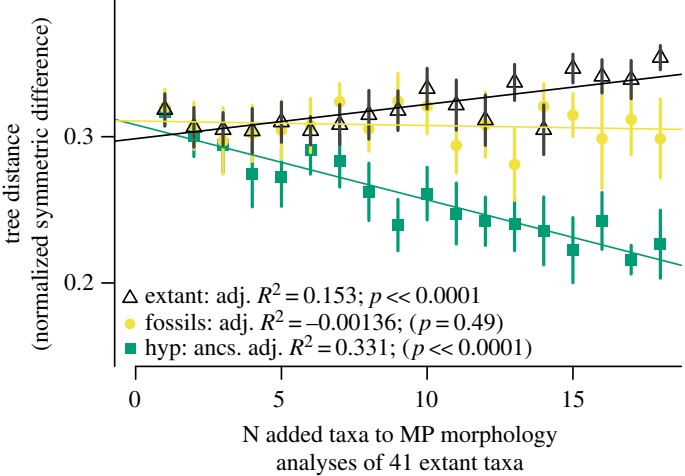

**Figure 7.** Normalized symmetric difference between the well-corroborated tree (figure 3) and strict consensuses generated by equal weights MP analysis of morphological data for 41 taxa in well-corroborated tree plus random combinations of 1–18 added extant species (triangles), fossils (circles), or hypothetical ancestors (squares), each sampling 18 replicates per number of added taxa on X-axis. Vertical error bars represent one standard deviation on either side of the mean (polygons). Adjusted R-squared and significance values based on linear models (lm) calculated for each pair of x and y axes in R.

Bayesian analysis of morphology only, 13 with MP and fossils, and 17 with either MP and hypothetical ancestors or Bayesian analysis of morphology including fossils. Implied weights in TNT [34,35] improved congruence substantially for morphology of extant taxa alone, slightly for the combined dataset, and not at all for DNA+indels for extant taxa alone (electronic supplementary material, figure S6). Using just morphological data from extant taxa, the number of shared partitions increased from six to 13 with implied weighting concavity ($k$) constants of 2, 4, 6 or 8 and dropped off to seven with $k$ at or over 32. For weighted MP analyses including fossils, the number of shared partitions increased from 13 to 16 when $k = 24$ and returned to 13 with $k$ at 128, 256 or 512. Weighted MP analyses of the combined dataset of living and fossil taxa increased the number of shared partitions from 26 to 27 for all $k$ values we explored from 2 to 512. Unlike the equally weighted analysis, implied weights consistently placed the fossils *Anagale*, *Pseudictops* and *Anagalopsis* with *Rhynchocyon* outside Euarchontoglires, resulting in 27 rather than 26 shared partitions with the well-corroborated tree. For DNA+indels of extant taxa alone, MP recovered 27 of 28 possible shared partitions under equal weighting and all $k$ values we explored (electronic supplementary material, figure S6).

The use of hypothetical ancestors as terminals in our dataset improved congruence, nearly tripling the number of well-corroborated groups from six in the topology based on extant taxa only to 17 (electronic supplementary material, figure S5). Inclusion of hypothetical ancestors in our morphology-only analysis recovers such groups as diprotodonts, placentals, euarchontoglires, glirids, sciuroids and *Aplodontia*, myodonts, and geomyoids. MP applied to morphological data including fossils also recovered diprotodonts and Eutheria (showing some Cretaceous non-placental eutherians in a polytomy with extant placentals), along with a monophyletic Glires, Rodentia, and Ctenohystrica, but did not recover myomorphs, sciuromorphs, or major groups within the former such as myodonts or geomyoids.

## 3.2. The evolution of rodents and lagomorphs

Our dataset supports the same basic phylogenetic structure for Mammalia and Glires (figure 4; electronic supplementary material, figure S2) as do independent studies of phylogenetic data that we did not sample (figure 3). For example, Placentalia is monophyletic; Glires, primates and *Tupaia* share a common ancestor to the exclusion of *Rhynchocyon*; rodents and lagomorphs are sister taxa; *Ochotona* is the sister taxon to Leporidae within Lagomorpha. Extant lagomorphs, in turn, comprise the crown radiation within Duplicidentata, which also includes early Eocene fossils such as *Gomphos*. The total clade including Rodentia is Simplicidentata and includes taxa such as *Rhombomylus* (unresolved in equally weighted MP; electronic supplementary material, figure S2), followed by fossils such as *Tribosphenomys* and *Paramys*, then crown Rodentia. Our Bayesian topology weakly supports the sciuromorph clade at the rodent root (figure 4).

The articulated skeleton of *Heliscomys* (figure 1) sheds light on the early evolution of the myomorph (mouse-related) clade. Cope [36] first described *Heliscomys vetus* based on a lower jaw with four teeth from the Oligocene Cedar Creek beds of Colorado, USA. Until now, the most complete, published material of *Heliscomys* consisted of cranial fragments [8]. Our skeleton (USNM PAL 720183) represents *H. ostranderi* and is preserved in a nodule collected from the Chadronian (latest Eocene) locality of Jenny's Pocket at Flagstaff Rim, Wyoming [8]. MicroCT scanning reveals a skull and jaws, associated vertebral column and proximal ribs, forelimbs and articulated manus, displaced but associated hindlimbs with articulated pes, a partial femur, and an innominate, and enabled us to code 182 of 219 morphological characters in our matrix.

Our analysis is the first based on a combined DNA+morphology matrix to place *Heliscomys* with extant geomyoid rodents near the base of the myomorph (mouse-related) radiation. It also confirms previous interpretations that *Heliscomys* is related to geomyoids [8,37]. Our data further render 'heteromyids' paraphyletic, a possibility implied by the interpretation of *Heliscomys* as ancestral to all living geomyoids [37] and by patterns of cranial vasculature in extant species [38]. Our most congruent phylogenetic analysis (figure 4) shows *Heliscomys* and the extant *Dipodomys* outside the remaining 'heteromyid'-grade taxa in our sample (*Heteromys*, *Perognathus* and *Chaetodipus*). All of our Bayesian and MP analyses support 'heteromyid' paraphyly as well as a terrestrial, ricochetal ancestral morphology for the two ecologically very different geomyoid groups alive today: fossorial geomyids and terrestrial 'heteromyids'. The ricochetal locomotion in extant 'heteromyids' is in fact a phylogenetically ancient feature of the geomyoid clade. In our taxon sample, *Heliscomys* is the earliest-diverging geomyoid, but shows an elongate metatarsus and some reduction of the lateral digits, particularly digit I which shows a proximal phalanx terminating before the distal end of the adjacent metatarsal II (figure 1). Our results also support the placement of *Castor* with geomyoids, consistent with Castorimorpha [39] and supported by (among other features) shared possession of an elongate infraorbital canal (figure 1).

Our study also includes data from the oldest articulated skeleton yet known for *Ischyromys* (figure 2; USNM PAL 617532), from the Duchesnean (Late Eocene) of West Canyon Creek, Wyoming [9]. *Ischyromys* is related to sciuromorphs. None of our Bayesian or MP topologies recovered a monophyletic 'Ischyromyoidea', a grade of fossil rodents sometimes classified together in the literature [40]. Instead, early Eocene 'ischyromyoids' such as *Paramys* fall outside crown Rodentia (figure 4). CT scanned petrosals of the late Eocene *Douglassciurus* along with data from its skull and skeleton [10] place this taxon close to *Ischyromys* among extant sciuromorphs. As originally described [10], *Douglassciurus* (then known as '*Protosciurus*' [41]) is indeed 'the oldest fossil squirrel', reconstructed at the base of Sciuridae in a polytomy with *Aplodontia*, sister to sciurids. Both *Ischyromys* and *Douglassciurus* resemble most extant sciurids, and differ from *Paramys*, in possessing features such as a lower molar entoconid anterior to the hypoconid, prominent medial curvature to the angular process of the jaw, and septae within the auditory bulla (figure 2). Contra previous literature [42], all *Ischyromys* crania sampled with microCT for this study (figure 2; electronic supplementary material, appendices S2 and S3) exhibit a groove for the stapedial artery traversing the middle ear.

## 4. Discussion

An evolutionary mechanism behind life's biodiversity predicts that patterns of relationship derived from independent sources of data should exhibit similarity [2] and that ever-larger samples analysed with appropriate methods should increase confidence in that similarity [1]. Our results bear out both expectations; optimal topologies derived from independent data are highly congruent with the well-corroborated tree, and congruence increases with larger samples from our DNA+indel+morphology dataset (figure 5) or more fossil taxa (figure 5). It is worth underscoring that the well-corroborated tree (figure 3), approximated by increased sampling from our DNA+indel+morphology dataset, is one of an astronomically huge number of possible topologies and was reconstructed from data [11,12,13,14] not used in any of our phylogenetic analyses (figures 4–7; electronic supplementary material, figures S2, S3 and S5).

Compared to extant taxa, fossils exist closer in geological time to the common ancestors hypothesized to exist by evolutionary theory. Therefore, fossils potentially have a greater resemblance than living taxa to these common ancestors [43,44]. When added to a given phylogenetic study in ever-increasing numbers, and assuming a general, positive correlation between elapsed time and the probability of character change [45], one would expect that the addition of fossil phenotypes to a given taxon

sample will increase similarity to the well-corroborated tree to a greater extent than addition of living phenotypes. This is similar to the observation that most methods reconstruct phylogenetic history more accurately with an abundance of short- rather than long-branched taxa [1,46,47]. Fossils can succeed in improving phylogenetic accuracy due to their capacity to break up long branches [43,44]. Stated differently, the living mammals sampled here are removed from the Mesozoic and Palaeogene common ancestors shared with other living taxa by many millions of years, meaning that there has been substantially more time for patterns of historical relationship to have been obscured by evolutionary change among living taxa than among Palaeogene fossils.

The morphology of hypothetical ancestors optimized on a well-corroborated tree derived from genomic data (figure 3) is independent of whatever one might find in the fossil record. Such ancestors comprise a novel means to incorporate an otherwise inaccessible, genomic signal into phylogenetic studies of fossils [16]. An optimal topology based on a morphological dataset that includes hypothetical ancestors represents a minimum path of change for anatomical characters given the well-corroborated topology by which such ancestors were estimated. Therefore, while of substantial utility to incorporate a genomic signal into studies of fossils that would otherwise lack such a signal [16], increased congruence with the addition of such hypothetical ancestors is in some sense trivial. We expect to observe increased resemblance towards that topology as more hypothetical ancestors are sampled. What is not trivial, and indeed a key finding of this study, is that increased sampling of fossils results in more topological congruence than increased sampling of extant taxa (figure 7). The morphology observed in fossils is independent of morphology of hypothetical ancestors estimated by optimizing characters on a well-corroborated tree of living taxa (figure 3). Addition of fossils and hypothetical ancestors known for just morphological characters has a net positive effect on congruence (figures 6 and 7), whereas addition of extant taxa decreases congruence (figure 7). We interpret this as evidence that palaeontology corroborates the morphological reconstructions of common ancestors hypothesized based on a well-corroborated topology. This is a quantitative example of the qualitative observation [48] that fossils mix phenotypic features intermediate between daughter species and the common ancestors they share. Accordingly, fossils help to constrain the morphological transformation series through which extant taxa have passed relative to that common ancestor [49].

Addition of living taxa generally has the effect of improving phylogenetic accuracy [50], not least because of the huge amounts of data typically available for them. However, with the important qualification that evolutionary rate can vary between and within lineages, living taxa known for morphology alone potentially have a greater handicap in phylogenetic reconstruction than most fossils or hypothetical ancestors; they are neither temporally close to geologically distant branching events, nor are they defined by morphological character states optimized on a well-corroborated tree. We do not claim that addition of morphological data from extant taxa necessarily decreases congruence, as indeed there is evidence that such addition can improve it [44]. Rather, our expectation is that, assuming some consistency of evolutionary rate across branches and over time, addition of morphological data from fossils to a given phylogenetic study should generally result in more congruence compared to the addition of morphological data from living taxa. Our results bear out this expectation (figure 7).

## 5. Conclusion

Our data support the interpretation, originally made based on much more limited and fragmentary fossil remains, that *Heliscomys* is a geomyoid, that 'heteromyids' are paraphyletic, and that extant geomyoids share a ricochetal common ancestor and are part of the myomorph radiation of Rodentia. Our analysis also supports the association of *Ischyromys* and *Douglassciurus* with extant sciuromorphs and the paraphyly of 'ischyromyoids'. Less decisively, our data also support the sciuromorph root favoured by an independent SINE dataset [11]. In this study, genomic data such as SINEs [11], microRNAs [12], introns [13] and RGCs [14] independently defined the well-corroborated tree and thus needed to be kept separate in order to be able to measure congruence in our dataset without circularity. However, an obvious step for future work would be to combine these datasets and thereby resolve further, high-level questions about mammalian phylogeny with a yet-larger pool of data, one which would further enable phylogenetic reconstructions of both living and fossil species [7,51]. More generally, and as predicted by evolution, our study confirms that distinct bodies of heritable data, such as morphology and DNA, converge towards a single mammalian tree of life. Even when reduced to hard tissues via fossilization, morphology positively contributes to identifying the tree-like pattern by which life has evolved.

Ethics. Our work involved no live animals and we were not required to complete an ethical assessment prior to conducting our research. Palaeontological fieldwork was carried out in accordance with national and regional laws.

Data accessibility. Our supplementary material section includes additional details on systematic and phylogenetic methods, figures S1–S7 and tables S1–S5. A downloadable morphological matrix and graphic documentation of all morphological character states are available in project 2769 on morphobank.org/permalink/?P2769 [23]. Four appendices are available via Dryad https://doi.org/10.5061/dryad.3840vd7 [24]. Electronic supplementary material, appendix S1 is an archive with DNA alignment, indels, combined data matrix, and hypothetical ancestors in nexus format. Electronic supplementary material, appendix S2 is a spreadsheet with morphological character edits. Electronic supplementary material, appendix S3 is a spreadsheet with genus-species names, museum numbers, and DNA accession numbers. Electronic supplementary material, appendix S4 provides optimal trees in nexus format.

Authors' contributions. R.J.A. conceived the project. The text was written by R.J.A. and M.R.S., with additional edits to text and figures from A.R. and R.J.E. R.J.A., R.J.E. and A.R. collected the data; R.J.E. found and collected USNM 617532 and USNM 720183 and made available further USNM specimens; R.J.A. undertook CT scans; R.J.A. and M.R.S. analysed the data and made the figures.

Competing interests. We have no competing interests.

Funding. Our work has been supported by the Department of Zoology, University of Cambridge, the University of Durham, and the Smithsonian Institution. A.R. and R.A. acknowledge support from a Claire Barnes graduate fellowship in the Department of Zoology, University of Cambridge.

Acknowledgements. We have many individuals to thank, in alphabetical order by institution, for helping to support our research. For access to key living and fossil specimens we thank Ewan St. John Smith (Cambridge), Doug Boyer and Arianna Harrington (Durham USA), Pip Brewer, Jerry Hooker and Roberto Portela Miguez (London), Pierre-Henri Fabre and Lionel Hautier (Montpellier), Meng Jin (New York), Roger Benson (Oxford), Anthony Herrel (Paris), Brian Kraatz (Pomona), Ornella Bertrand and Mary Silcox (Toronto), Cathrin Pfaff (Vienna) and Jessica Nakano and Jennifer Strotman (Washington). We are also grateful to the online resources at digimorph.org (Austin) and morphosource.org (Durham USA). For discussion of phylogenetic methods and software we thank Seraina Klopfstein (Basel), Joe Keating and Robert Sansom (Manchester) and Robin Beck (Salford). We thank Stuart Rankin, Jeffrey Salmond (Cambridge) and Caroline Willich (Cambridge, Ulm) for assistance with the University of Cambridge HPC cluster. We are grateful to Robert Sansom and two anonymous reviewers for their constructive comments on the manuscript.

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
