## [Reviewer comments · Royal Society Open Science]

Review History

RSOS-190387.R0 (Original submission)

Review form: Reviewer 1

Is the manuscript scientifically sound in its present form?

Yes

Are the interpretations and conclusions justified by the results?

Yes

Is the language acceptable?

Yes

Is it clear how to access all supporting data?

Yes

Do you have any ethical concerns with this paper?

No

Have you any concerns about statistical analyses in this paper?

No

Recommendation?

Accept with minor revision (please list in comments)

Comments to the Author(s)

This is an interesting paper tackling an important question that concerns many palaeontologists, and particularly those used to dealing with mammal data, given the recent history of publications casting doubt on the efficacy of, in particular, dental characters in mammal evolution. I recommend publication but have some suggestions that I think would improve the manuscript. The paper has two aspects. One is a characterisation and phylogenetic analysis of the geomyoid *Heliscomys*. On that I have nothing to criticise; the statements made are consistent with the results, the methodology for establishing the tree is sound, and the literature is fairly represented. The major aspect of this paper, though, is the test of congruence, and I have a few points to make here:

HYPOTHETICAL ANCESTORS

I am dubious about the utility of using hypothetical ancestors because the reasoning seems to me to be very circular. While I appreciate that the morphological data are independent of the molecular data used to define the well corroborated tree (as mentioned on lines 314-316), there is circularity in character state reconstructions.

The ancestral states are defined by applying a hypothetical model of character evolution to a phylogeny. Then, those ancestral states are used with a hypothetical model of character evolution (which might be the same or different) to determine a phylogeny, and the two phylogenies are compared. That inclusion of ancestors defined by a tree strengthen the congruence between the original and the estimated tree is surely inevitable, especially where the models concerned are the same. As long as both maximum parsimony and Bayesian models are giving a broadly similar topology to begin with, that will remain true even when the evolutionary model used to estimate ancestral states is different from the one that is used to estimate the tree. The authors do make concessions in this direction, calling the results "trivial", which I suppose is fair enough, but that just raises the question as to why they were tested in the first place.

ACCTTRAN AND DELTRAN

ACCTTRAN and DELTRAN also represent end-members of a spectrum, and both will tend to concentrate ambiguous state transitions to early or late branches, especially when portions of the anatomy are missing in several sequentially branching taxa.

If the results show little difference, then that is reassuring, and this is what the authors state – "Acctran and deltran generated similar results", lines 96-97. Were the ACCTTRAN matrices analysed at all, why were the DELTRAN datasets selected, and would it be possible to have a quick run-down of any key differences?

DETAILS OF CONGRUITY

The parts of the paper that discuss where conflict exists (lines 114-115, 124-125, etc) talk about the positions of Dermoptera and Primates. Given that these data sets are constructed to discuss the evolution of Glires, particularly rodents, the choice of morphological characters might not be amenable to properly elucidating the relationships among the euarchontan outgroup.

I am confused as to what the difference is between the processes that led to Figure 6 and Figure 7. For Figure 6, it starts with 60 extant taxa, and then add different numbers of fossils and HAs, and increasing numbers of both result in greater congruence in terms of both shared quartets and normalised symmetric difference.

For Figure 7, 19 taxa that are less well corroborated are removed (giving 41) and replaced with up

to 18 of the 18 other taxa, either extant, fossil, or HA. When this was done, fossils do not reduce normalised symmetric difference, but HAs still do. Indeed, the negative R2 value indicates that in fact the model is a marginally worse fit for the data than a horizontal line

Firstly, I don't follow why there are 18 sets that are constructed rather than 19; if the rationale is that there is exactly 1 set of 60 extant taxa, there should be 19 ways of having 41+18 extant taxa, with one of 19 dropped each time.

Secondly, the authors could do a lot more to explain why the results are so different with respect to the effect of adding fossils on congruence when the methods seem to be almost identical. I've been thinking about this, and the difference, as far as I can tell, is that in the first analysis, the trees include relatively unstable taxa (for a certain value of unstable). Including fossils there will perhaps introduce those ancestral states that can help to tie rogue taxa down more firmly, and so we might indeed expect that fossil taxa will aid in increasing congruence with independent data. In the second case, the 41 extant taxa that are included are already fairly well established, and are going to be more stable anyway. Adding more fossil taxa apparently has no effect on the congruence of these taxa with the well-corroborated tree. Perhaps this indicates that fossils are at their most useful in increasing congruence where there is phylogenetic uncertainty among extant forms. Here, too, the addition of extant taxa explicitly means adding in taxa whose position is less well corroborated, and as a result, it is perhaps not too surprising that the congruence values decrease. The authors state (lines 301-302) "We do not claim that addition of morphological data from extant taxa necessarily decreases congruence", and perhaps this is a way of explaining the results in that context, because the raw results certainly suggest that increasing extant taxon representation decreases congruence.

A relatively minor point on phrasing for Figure 7 - the sentence on lines 282-283 ending "increases sampling of fossils results in more topological congruence than increased sampling of extant taxa" gives an impression that could be misleading, suggesting that the addition of fossils improves congruence, when no such result was found. I appreciate that because the addition of extant taxa actively reduces congruence, and because of the results shown in Figure 6, it can be defended as "more topological congruence", but being explicit about the actual relationships in the text will avoid misunderstanding. When I looked at Figure 7, I had certainly not expected those values given the associated text.

I suggest that these sections would substantially benefit from rephrasing and the expansion of discussion as to why it is that one analysis suggests that fossils are beneficial as regards congruence and one very similar analysis finds that they have no effect.

Review form: Reviewer 2 (Michael Lee)

Is the manuscript scientifically sound in its present form?

Yes

Are the interpretations and conclusions justified by the results?

Yes

Is the language acceptable?

Yes

Is it clear how to access all supporting data?

Yes

Do you have any ethical concerns with this paper?

No

Have you any concerns about statistical analyses in this paper?

No

Recommendation?

Accept with minor revision (please list in comments)

Comments to the Author(s)

This paper presents novel information about the affinities of important rodent fossils, and also discusses important issues regarding congruence of phylogenetic datasets and the utility of morphological traits, especially skeletal anatomy. It is well-written and potentially of broad interest; it presents novel empirical data and the analyses are all well-executed.

There are no major issues that need to be addressed, but the authors may choose to discuss some of the comments below.

MAIN TEXT

1 >we used the "describetrees /xout=internal" command in PAUP 4.0a (22) to estimate states for our 219 morphological characters for each of the 29 internal nodes in the well-corroborated topology (Fig. 3).

Should be acknowledged that optimising traits this polytomous consensus tree might be an issue; the polytomy is most likely soft (uncertainty) but the optimisation treats it as hard (simultaneous speciation). This is problematic because character 1 might be congruent with the resolution of the polytomy in most-parsimonious topology A, and character 2 likewise for topology B, but for the consensus, both characters will be homoplasious across the polytomy (see Cladistics 12:139 etc). However, this bias works in the opposite direction to the positive results obtained - ie the reconstructed ancestors greatly improve phylogenetic signal even though the ancestral states reconstructed at the polytomies might be less from ideal - so if anything it might be said to improve confidence in the retrieved patterns.

2 >This tree was based on data noted above (9,10,11,12), independent of the DNA and morphological data used in our own phylogenetic analyses.

Largely independent, but maybe not totally? e.g. ref 10 uses the Dos Reis et al. nuclear and mt genes (not just introns) which presumably includes some of the loci used here? If so, please add qualifer.

3 The resampling processes (Results line 116+) really should be in the Methods. I understand why they are discussed in detail here, but at the very least, they should be briefly flagged/summarised in the Methods.

4 >Line 162: addition of fossils did not monotonically do so, but addition of 18 distinct sets of 18 fossils each did result in lower mean symmetric difference

I see no significant trend here! Maybe tone this down (strongly).

5 >line 163: In contrast, successive addition of randomized sets of up to 18 other extant taxa to

the 41-taxon set from our well-corroborated tree significantly increased conflict as measured by normalized symmetric difference with the well-corroborated tree

Presumably, these 18 taxa already had close relatives sequenced and thus included, and so contributed relatively little 'extra info' towards deeper-level relationships?

6 Implied weights analyses. This is probably for outside the scope of the present paper, but it would be interesting to compare implied weights (individual characters) in the morph only, and in the morph+DNA, analyses. The latter might better identify and automatically downweight homoplasious morph characters (see also Davalos et al. Syst Biol).

FIGURES

Figs 3 and 4 - please use genus names (or full names) rather than acronyms

Fig 5 - the morph has a bigger (and positive) impact when the molecular data set is small. The bigger impact is inevitable due to relative data set size, but this impact could (theoretically) been either helpful or unhelpful, given debates over the usefulness of morphology.

Fig 6 - colour key (ie same as fig 5) required.

Decision letter (RSOS-190387.R0)

29-May-2019

Dear Dr Asher

On behalf of the Editors, I am pleased to inform you that your Manuscript RSOS-190387 entitled "Congruence, fossils, and the evolutionary tree of rodents and lagomorphs" has been accepted for publication in Royal Society Open Science subject to minor revision in accordance with the referee suggestions. Please find the referees' comments at the end of this email.

The reviewers and handling editors have recommended publication, but also suggest some minor revisions to your manuscript. Therefore, I invite you to respond to the comments and revise your manuscript.

- Ethics statement

- Data accessibility

It is a condition of publication that all supporting data are made available either as supplementary information or preferably in a suitable permanent repository. The data accessibility section should state where the article's supporting data can be accessed. This section should also include details, where possible of where to access other relevant research materials such as statistical tools, protocols, software etc can be accessed. If the data has been deposited in an external repository this section should list the database, accession number and link to the DOI for all data from the article that has been made publicly available. Data sets that have been

deposited in an external repository and have a DOI should also be appropriately cited in the manuscript and included in the reference list.

If you wish to submit your supporting data or code to Dryad (<http://datadryad.org/>), or modify your current submission to dryad, please use the following link:
<http://datadryad.org/submit?journalID=RSOS&manu=RSOS-190387>

- **Competing interests**

- **Authors' contributions**

- **Acknowledgements**

- **Funding statement**

Because the schedule for publication is very tight, it is a condition of publication that you submit the revised version of your manuscript before 07-Jun-2019. Please note that the revision deadline will expire at 00.00am on this date. If you do not think you will be able to meet this date please let me know immediately.

When submitting your revised manuscript, you will be able to respond to the comments made by the referees and upload a file "Response to Referees" in "Section 6 - File Upload". You can use this

to document any changes you make to the original manuscript. In order to expedite the processing of the revised manuscript, please be as specific as possible in your response to the referees. We strongly recommend uploading two versions of your revised manuscript:

Kind regards,
Andrew Dunn
Royal Society Open Science Editorial Office

on behalf of Dr Robert Sansom (Associate Editor) and Kevin Padian (Subject Editor)
 openscience@royalsociety.org

Associate Editor Comments to Author (Dr Robert Sansom):

Associate Editor: 1

Comments to the Author:

Please take a look at the minor revisions suggested by both reviewers. Both raise some minor but important points for consideration, but I anticipate them being easily addressed in order to acceptance of publication. I also invite the authors to more carefully consider just how 'independent' the two datasets being compared are. Also can you explain in more detail what 'well corroborated' means in this instance (can it be defined or quantified)? To increase the impact and reach of your figures, I suggest adding more informative axis labels so that a reader can interpret them more easily e.g. "Proportion of phylogenetic signal recovered (shared quartets, normalized to reference tree)", but that is really up to your preferred style.

Reviewer comments to Author:

Reviewer: 1

Comments to the Author(s)

This is an interesting paper tackling an important question that concerns many palaeontologists, and particularly those used to dealing with mammal data, given the recent history of publications casting doubt on the efficacy of, in particular, dental characters in mammal evolution. I recommend publication but have some suggestions that I think would improve the manuscript. The paper has two aspects. One is a characterisation and phylogenetic analysis of the geomyoid *Heliscomys*. On that I have nothing to criticise; the statements made are consistent with the results, the methodology for establishing the tree is sound, and the literature is fairly represented. The major aspect of this paper, though, is the test of congruence, and I have a few points to make here:

HYPOTHETICAL ANCESTORS

I am dubious about the utility of using hypothetical ancestors because the reasoning seems to me to be very circular. While I appreciate that the morphological data are independent of the molecular data used to define the well corroborated tree (as mentioned on lines 314-316), there is circularity in character state reconstructions.

The ancestral states are defined by applying a hypothetical model of character evolution to a phylogeny. Then, those ancestral states are used with a hypothetical model of character evolution (which might be the same or different) to determine a phylogeny, and the two phylogenies are compared. That inclusion of ancestors defined by a tree strengthen the congruence between the original and the estimated tree is surely inevitable, especially where the models concerned are the same. As long as both maximum parsimony and Bayesian models are giving a broadly similar topology to begin with, that will remain true even when the evolutionary model used to estimate ancestral states is different from the one that is used to estimate the tree. The authors do make concessions in this direction, calling the results "trivial", which I suppose is fair enough, but that just raises the question as to why they were tested in the first place.

ACCTTRAN AND DELTRAN

ACCTTRAN and DELTRAN also represent end-members of a spectrum, and both will tend to concentrate ambiguous state transitions to early or late branches, especially when portions of the anatomy are missing in several sequentially branching taxa.

If the results show little difference, then that is reassuring, and this is what the authors state – “Acctran and deltran generated similar results”, lines 96-97. Were the ACCTTRAN matrices analysed at all, why were the DELTRAN datasets selected, and would it be possible to have a quick run-down of any key differences?

DETAILS OF CONGRUITY

The parts of the paper that discuss where conflict exists (lines 114-115, 124-125, etc) talk about the positions of Dermoptera and Primates. Given that these data sets are constructed to discuss the evolution of Glires, particularly rodents, the choice of morphological characters might not be amenable to properly elucidating the relationships among the euarchontan outgroup.

I am confused as to what the difference is between the processes that led to Figure 6 and Figure 7. For Figure 6, it starts with 60 extant taxa, and then add different numbers of fossils and HAs, and increasing numbers of both result in greater congruence in terms of both shared quartets and normalised symmetric difference.

For Figure 7, 19 taxa that are less well corroborated are removed (giving 41) and replaced with up to 18 of the 18 other taxa, either extant, fossil, or HA. When this was done, fossils do not reduce normalised symmetric difference, but HAs still do. Indeed, the negative R² value indicates that in fact the model is a marginally worse fit for the data than a horizontal line

Firstly, I don't follow why there are 18 sets that are constructed rather than 19; if the rationale is that there is exactly 1 set of 60 extant taxa, there should be 19 ways of having 41+18 extant taxa, with one of 19 dropped each time.

Secondly, the authors could do a lot more to explain why the results are so different with respect to the effect of adding fossils on congruence when the methods seem to be almost identical. I've been thinking about this, and the difference, as far as I can tell, is that in the first analysis, the trees include relatively unstable taxa (for a certain value of unstable). Including fossils there will perhaps introduce those ancestral states that can help to tie rogue taxa down more firmly, and so we might indeed expect that fossil taxa will aid in increasing congruence with independent data. In the second case, the 41 extant taxa that are included are already fairly well established, and are going to be more stable anyway. Adding more fossil taxa apparently has no effect on the congruence of these taxa with the well-corroborated tree. Perhaps this indicates that fossils are at their most useful in increasing congruence where there is phylogenetic uncertainty among extant forms. Here, too, the addition of extant taxa explicitly means adding in taxa whose position is less well corroborated, and as a result, it is perhaps not too surprising that the congruence values decrease. The authors state (lines 301-302) “We do not claim that addition of morphological data from extant taxa necessarily decreases congruence”, and perhaps this is a way of explaining the results in that context, because the raw results certainly suggest that increasing extant taxon representation decreases congruence.

A relatively minor point on phrasing for Figure 7 – the sentence on lines 282-283 ending “increases sampling of fossils results in more topological congruence than increased sampling of extant taxa” gives an impression that could be misleading, suggesting that the addition of fossils improves congruence, when no such result was found. I appreciate that because the addition of extant taxa actively reduces congruence, and because of the results shown in Figure 6, it can be defended as “more topological congruence”, but being explicit about the actual relationships in the text will avoid misunderstanding. When I looked at Figure 7, I had certainly not expected those values given the associated text.

I suggest that these sections would substantially benefit from rephrasing and the expansion of discussion as to why it is that one analysis suggests that fossils are beneficial as regards congruence and one very similar analysis finds that they have no effect.

Reviewer: 2

Comments to the Author(s)

This paper presents novel information about the affinities of important rodent fossils, and also discusses important issues regarding congruence of phylogenetic datasets and the utility of morphological traits, especially skeletal anatomy. It is well-written and potentially of broad interest; it presents novel empirical data and the analyses are all well-executed.

There are no major issues that need to be addressed, but the authors may choose to discuss some of the comments below.

MAIN TEXT

1 >we used the "describetrees /xout=internal" command in PAUP 4.0a (22) to estimate states for our 219 morphological characters for each of the 29 internal nodes in the well-corroborated topology (Fig. 3).

Should be acknowledged that optimising traits this polytomous consensus tree might be an issue; the polytomy is most likely soft (uncertainty) but the optimisation treats it as hard (simultaneous speciation). This is problematic because character 1 might be congruent with the resolution of the polytomy in most-parsimonious topology A, and character 2 likewise for topology B, but for the consensus, both characters will be homoplasious across the polytomy (see Cladistics 12:139 etc). However, this bias works in the opposite direction to the positive results obtained - ie the reconstructed ancestors greatly improve phylogenetic signal even though the ancestral states reconstructed at the polytomies might be less from ideal - so if anything it might be said to improve confidence in the retrieved patterns.

2 >This tree was based on data noted above (9,10,11,12), independent of the DNA and morphological data used in our own phylogenetic analyses.

Largely independent, but maybe not totally? e.g. ref 10 uses the Dos Reis et al. nuclear and mt genes (not just introns) which presumably includes some of the loci used here? If so, please add qualifer.

3 The resampling processes (Results line 116+) really should be in the Methods. I understand why they are discussed in detail here, but at the very least, they should be briefly flagged/summarised in the Methods.

4 >Line 162: addition of fossils did not monotonically do so, but addition of 18 distinct sets of 18 fossils each did result in lower mean symmetric difference

I see no significant trend here! Maybe tone this down (strongly).

5 >line 163: In contrast, successive addition of randomized sets of up to 18 other extant taxa to the 41-taxon set from our well-corroborated tree significantly increased conflict as measured by normalized symmetric difference with the well-corroborated tree

Presumably, these 18 taxa already had close relatives sequenced and thus included, and so contributed relatively little 'extra info' towards deeper-level relationships?

6 Implied weights analyses. This is probably for outside the scope of the present paper, but it would be interesting to compare implied weights (individual characters) in the morph only, and in the morph+DNA, analyses. The latter might better identify and automatically downweight homoplasious morph characters (see also Davalos et al. Syst Biol).

FIGURES

Figs 3 and 4 - please use genus names (or full names) rather than acronyms

Fig 5 - the morph has a bigger (and positive) impact when the molecular data set is small. The bigger impact is inevitable due to relative data set size, but this impact could (theoretically) been either helpful or unhelpful, given debates over the usefulness of morphology.

Fig 6 - colour key (ie same as fig 5) required.

Author's Response to Decision Letter for (RSOS-190387.R0)

See Appendix A.

Decision letter (RSOS-190387.R1)

19-Jun-2019

Dear Dr Asher,

I am pleased to inform you that your manuscript entitled "Congruence, fossils, and the evolutionary tree of rodents and lagomorphs" is now accepted for publication in Royal Society Open Science.

Kind regards,
Lianne Parkhouse
Royal Society Open Science

on behalf of Kevin Padian (Subject Editor)
openscience@royalsociety.org

Appendix A

Dear Dr. Sansom (Rob),

Many thanks for your constructive comments & for handling those of the reviewers. These have enabled a number of improvements, as summarized point-by-point below. Beyond the specifics requested by you & the reviewers, we've added more background citations (#3 below) and have clarified our methods for comparing topologies (from line 124). I hope all of this is to your satisfaction and again my coauthors and I are very grateful for your time.

Sincere regards, Robert Asher

Associate Editor Comments to Author (Dr Robert Sansom):

Associate Editor: 1

Comments to the Author:

Please take a look at the minor revisions suggested by both reviewers. Both raise some minor but important points for consideration, but I anticipate them being easily addressed in order to acceptance of publication. I also invite the authors to more carefully consider just how 'independent' the two datasets being compared are. Also can you explain in more detail what 'well corroborated' means in this instance (can it be defined or quantified)? To 1. We've added text to the last paragraph of the introduction to clarify the independent studies behind the topology in Fig. 3, as well as our understanding of "well corroborated" (from line 49):

"... a well-corroborated tree (Fig. 3), derived from SINEs (), microRNAs (: their fig. S5), introns (), and rare genomic changes (), data which played no role in our own original dataset. This tree is furthermore consistent with an analysis of ultraconserved elements from over 3700 nuclear loci (), which shows less than 0.002% (8/3700) overlap with the eight nuclear genes in our original alignment. By "well-corroborated" we mean that one or an extremely small number of mutually consistent topologies, out of an astronomically huge number of possibilities, result from phylogenetic optimality criteria applied to these datasets."

increase the impact and reach of your figures, I suggest adding more informative axis labels so that a reader can interpret them more easily e.g. "Proportion of phylogenetic signal recovered (shared quartets, normalized to reference tree)", but that is really up to your preferred style.

2. We have modified the Y axis labels of the relevant main (5-7) & supplementary (S1, S4, S6, S7) figures to include reference to the well-corroborated tree and similarity metric.

Reviewer comments to Author:

Reviewer: 1

Comments to the Author(s)

This is an interesting paper tackling an important question that concerns many palaeontologists, and particularly those used to dealing with mammal data, given the recent history of publications casting doubt on the efficacy of, in particular, dental characters in mammal evolution. I recommend publication but have some suggestions that I think would improve the manuscript.

3. We have added additional background to the first introductory paragraph (from line 39), better acknowledging the literature concerning phylogenetic information content in fossils.

The paper has two aspects. One is a characterisation and phylogenetic analysis of the geomyoid *Heliscomys*. On that I have nothing to criticise; the statements made are consistent with the results, the methodology for establishing the tree is sound, and the literature is fairly represented.

The major aspect of this paper, though, is the test of congruence, and I have a few points to make here:

HYPOTHETICAL ANCESTORS

I am dubious about the utility of using hypothetical ancestors because the reasoning seems to me to be very circular. While I appreciate that the morphological data are independent of the molecular data used to define the well corroborated tree (as mentioned on lines 314-316), there is circularity in character state reconstructions.

The ancestral states are defined by applying a hypothetical model of character evolution to a phylogeny. Then, those ancestral states are used with a hypothetical model of character evolution (which might be the same or different) to determine a phylogeny, and the two phylogenies are compared. That inclusion of ancestors defined by a tree strengthen the congruence between the original and the estimated tree is surely inevitable, especially where the models concerned are the same. As long as both maximum parsimony and Bayesian models are giving a broadly similar topology to begin with, that will remain true even when the evolutionary model used to estimate ancestral states is different from the one that is used to estimate the tree. The authors do make concessions in this direction, calling the results “trivial”, which I suppose is fair enough, but that just raises the question as to why they were tested in the first place.

4. Our purpose in adding hypothetical ancestors to our phylogenetic analysis is not primarily the resulting phylogeny (our Fig. S5c, not figured in the main text), but rather to compare iterative addition of such hypothetical ancestors vs. real fossils and living taxa (our figs. 6-7). We've found that addition of fossils and hypothetical ancestors both increase congruence (up to 41 fossils & 28 ancestors added to analysis of 60 extant taxa, Fig. 6) whereas addition of living taxa (up to 18 taxa added to analysis of 41 other living taxa, Fig. 7) decreases congruence. We're obliged to note that the smaller comparison in Fig. 7 shows fossils without a significant improvement in normalized symmetric difference from 1 to 18 taxa (see points 7 and 11, below and our new supplementary Fig. S4), but even so, and unlike addition of extant taxa, addition of fossils does not negatively impact congruence (and again the larger, 60+41 fossil taxon comparison in Fig. 6 improves congruence). The method of Beck & Baillie (who introduced the idea of adding hypothetical ancestors in paleosystematics, not us) was not intended to test phylogenetic hypotheses based on genomic data, but only as an additional tool for paleontologists to include signal from otherwise inaccessible genomic data in their analyses. We've further elaborated our text to help clarify this in the paragraph starting on line 99:

"[Beck & Baillie 2018] was not a test of evolution per se, or a test of the topology proposed by Meredith et al (), but it did provide a means by which to increase congruence of topologies derived from fossils with topologies based on living taxa sampled for DNA.

The method of Beck & Baille () thus comprises a means to include the phylogenetic information content from modern taxa in an analysis of fossils, without directly having to sample non-fossilizable data, such as DNA."

As the reviewer noted, our text also explains this point from line 311:

"... while of substantial utility to incorporate a genomic signal into studies of fossils that would otherwise lack such a signal (), increased congruence with the addition of such hypothetical ancestors is in some sense trivial. We expect to observe increased resemblance towards that topology as more hypothetical ancestors are sampled. What is not trivial, and indeed a key finding of this study, is that increased sampling of fossils results in more topological congruence than increased sampling of extant taxa (Figs. 7, S4)"

ACCTTRAN AND DELTRAN

ACCTTRAN and DELTRAN also represent end-members of a spectrum, and both will tend to concentrate ambiguous state transitions to early or late branches, especially when portions of the anatomy are missing in several sequentially branching taxa.

If the results show little difference, then that is reassuring, and this is what the authors state – “Accttran and deltran generated similar results”, lines 96-97. Were the ACCTTRAN matrices analysed at all, why were the DELTRAN datasets selected, and would it be possible to have a quick run-down of any key differences?

5. We have added supplementary figure S1 (as mentioned e.g., on line 112), showing the similar pattern of increased congruence with the addition of hypothetical ancestors assuming both acctran and deltran optimizations. We have also added further detail to our methods to calculate topological similarity (from line 124).

DETAILS OF CONGRUITY

The parts of the paper that discuss where conflict exists (lines 114-115, 124-125, etc) talk about the positions of Dermoptera and Primates. Given that these data sets are constructed to discuss the evolution of Glires, particularly rodents, the choice of morphological characters might not be amenable to properly elucidating the relationships among the euarchontan outgroup.

6. We agree. We mentioned the variable position of these taxa only because it was a conspicuous difference between our MP and Bayesian results, potentially of further interest to subsequent methodological studies.

I am confused as to what the difference is between the processes that led to Figure 6 and Figure 7.

7. Fig. 6 shows added fossils and ancestors to our dataset including all extant taxa. In Fig. 7 we sought to compare the addition of fossils and ancestors to addition of living taxa, and thus we had to start with a smaller set of extant taxa in order to randomly subsample other living taxa. We have now added text (from line 182) clarifying this point:

"In order to compare the addition of fossils and hypothetical ancestors to addition of living taxa, we subsampled from our set of 60 extant taxa."

For Figure 6, it starts with 60 extant taxa, and then add different numbers of fossils and HAs, and increasing numbers of both result in greater congruence in terms of both shared quartets and normalised symmetric difference.

For Figure 7, 19 taxa that are less well corroborated are removed (giving 41) and replaced with up to 18 of the 18 other taxa, either extant, fossil, or HA. When this was done, fossils do not reduce normalised symmetric difference, but HAs still do. Indeed, the negative R2 value indicates that in fact the model is a marginally worse fit for the data than a horizontal line

8. The line for 1 to 18 fossils added to 41 living taxa (Fig. 7) shows a non-significant fit (and we left our root taxon, *Didelphis*, present in all analyses). Moreover, even though the trendline lacks a positive slope, the mean number of shared partitions is highest for subsamples of 18 vs. smaller numbers of added fossil taxa (now evident in the new Fig. S4). We have changed our text to point this out (from line 191). More importantly, the larger sample of all living taxa plus up to 41 fossils or 28 ancestors (Fig. 6) shows significant improvements in congruence as both categories of taxa are added to our analysis.

Firstly, I don't follow why there are 18 sets that are constructed rather than 19; if the rationale is that there is exactly 1 set of 60 extant taxa, there should be 19 ways of having 41+18 extant taxa, with one of 19 dropped each time.

9. Yes for 18 that is correct, whereas there is only one set of 41+19 taxa (=60, our total sample of living taxa). In order to construct Y-axis error bars, we needed multiple, randomly chosen sets of added taxa, so the X-axis of Fig. 7 shows a maximum of 18 added taxa.

Secondly, the authors could do a lot more to explain why the results are so different with respect to the effect of adding fossils on congruence when the methods seem to be almost identical. I've been thinking about this, and the difference, as far as I can tell, is that in the first analysis, the trees include relatively unstable taxa (for a certain value of unstable). Including fossils there will perhaps introduce those ancestral states that can help to tie rogue taxa down more firmly, and so we might indeed expect that fossil taxa will aid in increasing congruence with independent data.

10. The phylogenetic methods in reconstructing fossils, ancestors, or living taxa are the same. The reason for the different behavior in terms of increased congruence, we believe, is that fossils are better than living taxa at approximating the morphology hypothesized to exist by optimizing characters on the well-corroborated tree. We have explained this from line 292:

" Compared to extant taxa, fossils exist closer in geological time to the common ancestors hypothesized to exist by evolutionary theory. Therefore, fossils potentially have a greater resemblance than living taxa to these common ancestors (). When added to a given phylogenetic study in ever-increasing numbers, and assuming a general, positive correlation between time and the probability of character change (), one would expect that the addition of fossil phenotypes to a given taxon sample will increase similarity to the well-corroborated tree to a greater extent than addition of living phenotypes."

and from line 330,

"...living taxa known for morphology alone potentially have a greater handicap in phylogenetic reconstruction than most fossils or hypothetical ancestors; they are neither temporally close to geologically distant branching events, nor are they defined by morphological character states optimized on a well-corroborated tree"

In the second case, the 41 extant taxa that are included are already fairly well established, and are going to be more stable anyway. Adding more fossil taxa apparently has no effect on the congruence of these taxa with the well-corroborated tree.

11. As outlined in response #7 above, adding fossils significantly improves congruence (Fig. 6). Only in the smaller analysis with 41 subsampled living taxa plus up to 18 fossils (Fig. 7) is the effect of improved congruence with added fossils masked by the reduced sample size and by scaling to normalized symmetric difference in Fig. 7, which nonetheless shows that congruence with N added fossils is still better than N added living taxa. As noted above (#7), we have added text and supplementary figure S4 to clarify this result.

Perhaps this indicates that fossils are at their most useful in increasing congruence where there is phylogenetic uncertainty among extant forms. Here, too, the addition of extant taxa explicitly means adding in taxa whose position is less well corroborated, and as a result, it is perhaps not too surprising that the congruence values decrease. The authors state (lines 301-302) “We do not claim that addition of morphological data from extant taxa necessarily decreases congruence”, and perhaps this is a way of explaining the results in that context, because the raw results certainly suggest that increasing extant taxon representation decreases congruence.

A relatively minor point on phrasing for Figure 7 – the sentence on lines 282-283 ending “increases sampling of fossils results in more topological congruence than increased sampling of extant taxa” gives an impression that could be misleading, suggesting that the addition of fossils improves congruence, when no such result was found.

I appreciate that because the addition of extant taxa actively reduces congruence, and because of the results shown in Figure 6, it can be defended as “more topological congruence”, but being explicit about the actual relationships in the text will avoid misunderstanding. When I looked at Figure 7, I had certainly not expected those values given the associated text.

I suggest that these sections would substantially benefit from rephrasing and the expansion of discussion as to why it is that one analysis suggests that fossils are beneficial as regards congruence and one very similar analysis finds that they have no effect.

12. Addition of fossils *does* have an effect; see points #7 and #11 and our added supplementary figure S4. Comparing addition of extant taxa vs. fossils and the data represented in Fig. 7 and S4, the sentence “increased sampling of fossils results in more topological congruence than increased sampling of extant taxa” is true as written.

Reviewer: 2

Comments to the Author(s)

This paper presents novel information about the affinities of important rodent fossils, and also discusses important issues regarding congruence of phylogenetic datasets and the utility of morphological traits, especially skeletal anatomy. It is well-written and potentially of broad interest; it presents novel empirical data and the analyses are all well-executed.

There are no major issues that need to be addressed, but the authors may choose to discuss some of the comments below.

MAIN TEXT

1 >we used the "describetrees /xout=internal" command in PAUP 4.0a (22) to estimate states for our 219 morphological characters for each of the 29 internal nodes in the well-corroborated topology (Fig. 3).

Should be acknowledged that optimising traits this polytomous consensus tree might be an issue; the polytomy is most likely soft (uncertainty) but the optimisation treats it as hard (simultaneous speciation). This is problematic because character 1 might be congruent with the resolution of the polytomy in most-parsimonious topology A, and character 2 likewise for topology B, but for the consensus, both characters will be homoplasious across the polytomy (see *Cladistics* 12:139 etc). However, this bias works in the opposite direction to the positive results obtained - ie the reconstructed ancestors greatly improve phylogenetic signal even though the ancestral states reconstructed at the polytomies might be less from ideal - so if anything it might be said to improve confidence in the retrieved patterns.

13. We agree this is an important point to mention, ideally to be addressed in future analyses of ancestral character state reconstruction. We've briefly noted the issue and added reference to Coddington & Scharff 1996 on line 116.

2 >This tree was based on data noted above (9,10,11,12), independent of the DNA and morphological data used in our own phylogenetic analyses. Largely independent, but maybe not totally? e.g. ref 10 uses the Dos Reis et al. nuclear and mt genes (not just introns) which presumably includes some of the loci used here? If so, please add qualifer.

14. We refer to the microRNA-derived topology published by Tarver et al. (2016: their fig. S5), who improved on the genomic sample of Dos Reis et al. and did not use the latter's nuclear markers as a basis for the well corroborated tree. We have added "microRNAs" and "their fig. S5" after our citation of Tarver et al. on line 50.

3 The resampling processes (Results line 116+) really should be in the Methods. I understand why they are discussed in detail here, but at the very least, they should be briefly flagged/summarised in the Methods.

15. We have added a final paragraph to the end of Methods from line 131, as requested: "We evaluated our hypotheses on the phylogenetic information content of hard-tissue data and fossils through a series of resampling experiments. These iteratively increased the number of sampled DNA characters with and without morphology, and also iteratively increased the number of sampled fossils, hypothetical ancestors, and sub-sampled living taxa, as detailed below."

4 >Line 162: addition of fossils did not monotonically do so, but addition of 18 distinct sets of 18 fossils each did result in lower mean symmetric difference

I see no significant trend here! Maybe tone this down (strongly).

16. See points 7, 8, 11, 12, above.

5 >line 163: In contrast, successive addition of randomized sets of up to 18 other extant taxa to the 41-taxon set from our well-corroborated tree significantly increased conflict as measured by normalized symmetric difference with the well-corroborated tree

Presumably, these 18 taxa already had close relatives sequenced and thus included, and so contributed relatively little 'extra info' towards deeper-level relationships?

17. We agree that degree of relatedness (represented by branch lengths) across subsampled taxa is certainly a worthwhile point for further exploration. For now, we've noted in our revision (from line 195) that taxa not among those with well-corroborated affinities (Fig. 3), and randomly chosen in iterations from one to 18 taxa (Figs. 7, S4), included both relatively long- and short-branched taxa.

6 Implied weights analyses. This is probably for outside the scope of the present paper, but it would be interesting to compare implied weights (individual characters) in the morph only, and in the morph+DNA, analyses. The latter might better identify and automatically downweight homoplasious morph characters (see also Davalos et al. Syst Biol).

18. We agree this is possible but would prefer to leave this issue for exploration in a further study.

FIGURES

Figs 3 and 4 - please use genus names (or full names) rather than acronyms

19. OK; all figured taxa are now shown with full genus names.

Fig 5 - the morph has a bigger (and positive) impact when the molecular data set is small. The bigger impact is inevitable due to relative data set size, but this impact could (theoretically) been either helpful or unhelpful, given debates over the usefulness of morphology.

20. We agree that large datasets can in theory dominate smaller ones, but given the large numbers of invariant sites in DNA alignments, sizes of DNA vs. morph character sets are not directly comparable.

Fig 6 - colour key (ie same as fig 5) required.

21. OK